# Viral Pneumonia during the COVID-19 Pandemic, 2019–2021 Evoking Needs for SARS-CoV-2 and Additional Vaccinations

**DOI:** 10.3390/vaccines11050905

**Published:** 2023-04-27

**Authors:** Sheng-Chieh Lin, Hsiao-Chin Wang, Wen-Chuan Lin, Yung-Ting Kuo, Yi-Hsiang Hsu, Yin-Tai Tsai, Shou-Cheng Lu, Yuan-Hung Wang, Shih-Yen Chen

**Affiliations:** 1Division of Allergy, Asthma, and Immunology, Department of Pediatrics, Shuang Ho Hospital, Taipei Medical University, Taipei 23561, Taiwan; 2Department of Pediatrics, School of Medicine, College of Medicine, Taipei Medical University, Taipei 11031, Taiwan; 3Division of Neonatology, Department of Pediatrics, Shuang Ho Hospital, Taipei Medical University, Taipei 23561, Taiwan; 4Division of Pediatric Infectious Diseases, Department of Pediatrics, Shuang Ho Hospital, Taipei Medical University, Taipei 23561, Taiwan; 5Division of Pediatric Neurology, Department of Pediatrics, Shuang Ho Hospital, Taipei Medical University, Taipei 23561, Taiwan; 6Beth Israel Deaconess Medical Center, Harvard Medical School, Boston, MA 02215, USA; 7Broad Institute of MIT and Harvard, Cambridge, MA 02142, USA; 8Department of Medicine Laboratory, Shuang Ho Hospital, Taipei Medical University, New Taipei City 23561, Taiwan; 9Graduate Institute of Clinical Medicine, College of Medicine, Taipei Medical University, Taipei 11031, Taiwan; 10Department of Medical Research, Shuang Ho Hospital, Taipei Medical University, New Taipei City 23561, Taiwan; 11Division of Pediatric Gastroenterology, Department of Pediatrics, Shuang Ho Hospital, Taipei Medical University, Taipei 23561, Taiwan; 12TMU Research Center for Digestive Medicine, Taipei Medical University, Taipei 11031, Taiwan

**Keywords:** viral pneumonia, COVID-19, SARS-CoV-2, vaccination

## Abstract

Coronaviruses can cause pneumonia, with clinical symptoms that may be similar to the symptoms of other viral pneumonias. To our knowledge, there have been no reports regarding cases of pneumonia caused by coronaviruses and other viruses among hospitalized patients in the past 3 years before and during coronavirus disease 2019 (COVID-19). Here, we analysed the causes of viral pneumonia among hospitalized patients during the coronavirus disease 2019 (COVID-19) pandemic (2019–2021). Between September 2019 and April 2021, patients hospitalized at Shuang Ho Hospital in north Taiwan with a diagnosis of pneumonia were enrolled in this study. Age, sex, onset date, and season of occurrence were recorded. Respiratory tract pathogens were identified with molecular detection using the FilmArray^®^ platform from nasopharyngeal swabs. In total, 1147 patients (128 patients aged <18 years and 1019 patients aged ≥18 years) with pneumonia and identified respiratory tract pathogens were assessed. Among the 128 children with pneumonia, the dominant viral respiratory pathogen was rhinovirus (24.2%), followed by respiratory syncytial virus (RSV; 22.7%), parainfluenza virus (1 + 2 + 3 + 4) (17.2%), adenovirus (12.5%), metapneumovirus (9.4%), coronavirus (1.6%), and influenza virus (A + B) (1.6%). Among the 1019 adults with pneumonia, the dominant viral respiratory pathogen was rhinovirus (5.0%), followed by RSV (2.0%), coronavirus (2.0%), metapneumovirus (1.5%), parainfluenza virus (1 + 2 + 3 + 4) (1.1%), adenovirus (0.7%), and influenza virus (A + B) (0%). From 2019–2021, older patients (aged >65 years) with pneumonia tested positive for coronavirus most commonly in autumn. Coronavirus was not detected during summer in children or adults. Among children aged 0–6 years, RSV was the most common viral pathogen, and RSV infection occurred most often in autumn. Metapneumovirus infection occurred most often in spring in both children and adults. In contrast, influenza virus was not detected in patients with pneumonia in any season among children or adults from January 2020 to April 2021. Among all patients with pneumonia, the most common viral pathogens were rhinovirus in spring, adenovirus and rhinovirus in summer, RSV and rhinovirus in autumn, and parainfluenza virus in winter. Among children aged 0–6 years, RSV, rhinovirus, and adenovirus were detected in all seasons during the study period. In conclusion, the proportion of pneumonia cases caused by a viral pathogen was higher in children than the proportion in adults. The COVID-19 pandemic period evoked a need for SARS-CoV-2 (severe acute respiratory disease coronavirus 2) vaccination to prevent the severe complications of COVID-19. However, other viruses were also found. Vaccines for influenza were clinically applied. Active vaccines for other viral pathogens such as RSV, rhinovirus, metapneuomoccus, parainfluenza, and adenovirus may need to be developed for special groups in the future.

## 1. Introduction

Pneumonia is a common airway disease worldwide, and viral pneumonia is common among hospitalized patients. Viral pneumonia is usually noted in young children, and it can easily lead to hospitalization in infants. Moreover, viral pneumonia exacerbates the risk and severity of bacterial pneumonia, and it causes co-infections. The most common pathogens that cause viral pneumonia in children are rhinovirus, respiratory syncytial virus (RSV), influenza virus, parainfluenza virus, metapneumovirus, and coronavirus [1]. Clinical manifestations of viral pneumonia include fever, coughing, appetite loss, and fatigue [2]. Viral pneumonia is observed mostly during particular seasons or during an epidemic. However, the clinical course is longer than the course of bacterial pneumonia. Viral pneumonia usually does not involve obvious leucocytosis, elevated C-reactive protein levels, or elevated serum procalcitonin levels [1,2]. Chest X-rays also show bilateral or interstitial infiltrates in patients with viral pneumonia [2]. In adolescents and adults, pneumonia is usually caused by bacteria. However, the virus that fuelled the coronavirus disease 2019 (COVID-19) pandemic, severe acute respiratory disease coronavirus 2 (SARS-CoV-2), can also cause pneumonia [3]. Risk factors for severe disease include older age, male sex, and comorbidities. Approximately 15–30% of hospitalized patients develop COVID-19-associated acute respiratory distress syndrome [3]. However, to our knowledge, there have been no reports regarding cases of pneumonia caused by coronaviruses and other viruses among hospitalized patients in the past 3 years. Here, we analysed the causes of viral pneumonia among hospitalized patients at a medical centre in northern Taiwan during the COVID-19 pandemic (2019–2021). 

## 2. Patients and Methods

### 2.1. Patients

Between September 2019 and April 2021, patients hospitalized at Shuang Ho Hospital in north Taiwan with a diagnosis of pneumonia caused by an unknown pathogen, along with obvious pneumonia on lung imaging, were enrolled in this study (Table 1). Pneumonia pathogens were identified with data mining from the department of laboratory medicine of Shuang Ho hospital based on molecular PCR (polymerase chain reaction) detection methods from a nasopharyngeal swab via the FilmArray^®^ platform (BioFire Diagnostics, Salt Lake City, UT, USA). The general detection method process is as described below.

Clinical manifestations included coughing (dry or productive), other respiratory tract symptoms, or a fever. Nasopharyngeal swabs were collected, and demographic data (age, sex, onset date, and season of occurrence) were recorded. In Taiwan, spring extends from March through May, summer extends from June through August, autumn extends from September through November, and winter extends from December through February. All patients with a previous SARS-CoV-2 infection were excluded. The study was approved by Taipei Medical University Joint Institutional Review Board (TMUJIRB) No. N202103153.

### 2.2. Identification of Viral Infections, 2019–2021

#### 2.2.1. Operation Process in the FilmArray System

The push rod inside the instrument pushed the samples from the original position of the column into the detection bag. The samples were mechanically squeezed by the instrument and collided with the white ceramic beads that produced cell lysates and released nucleic acid substances. These nucleic acid substances were adsorbed by black magnetic beads that were located in the detection bag. Then the 1–3 slots in the bag were squeezed back and forth several times to purify the nucleic acid. The instrument injected buffer from the back-end column to extract the purified nucleic acid, and then entered the detection bag to do the next-step PCR procedure.

#### 2.2.2. Polymerase Chain Reaction

The PCR primers/probes and conditions for identification of viral pathogens for lower respiratory tract infections were as reported previously. [4,5,6,7] The PCR reaction was started after synthesis of the first strand of cDNA and the PCR condition depended on the Tm of each primer according to manufacturer’s instructions. 

## 3. Results

From September 2019 to April 2021, 1147 patients with pneumonia based on molecular detection from a nasopharyngeal swab via the FilmArray^®^ platform were enrolled in the study. In total, 128 patients were aged <18 years, whereas 1019 patients were aged ≥18 years. Among the 128 paediatric patients (77 boys and 51 girls), the mean age was 3.65 years (standard deviation: 4.27); the majority of patients (100 children, 78%) were aged <6 years. Among these children, the most common pathogen was rhinovirus, followed by RSV, parainfluenza virus (1 + 2 + 3 + 4), adenovirus, metapneumovirus, *Mycoplasma pneumoniae*, influenza virus (A + B), and coronavirus. In descending order, the distribution of respiratory tract pathogens was rhinovirus (24.2%), RSV (22.7%), adenovirus (12.5%), metapneumovirus (9.4%), parainfluenza virus type 3 (7.0%), *M. pneumoniae* (5.5%), parainfluenza virus type 4 (5.5%), parainfluenza virus type 1 (3.9%), coronavirus (1.6%), influenza A (1.6%), and parainfluenza virus type 2 (0.8%) (Table 1). Among the 1019 adult patients (684 men and 335 women), the mean age was 70.9 years (standard deviation: 16.87); the majority of patients (732 adults, 72%) were aged >65 years. The most common pathogen was rhinovirus, followed by RSV, coronavirus, metapneumovirus, parainfluenza virus (1 + 2 + 3 + 4), and adenovirus. In descending order, the distribution of respiratory tract pathogens was rhinovirus (5.0%) followed by RSV (2.0%), coronavirus (2.0%), metapneumovirus (1.5%), adenovirus (0.7%), parainfluenza virus type 4 (0.6%), parainfluenza virus type 3 (0.4%), parainfluenza virus type 1 (0.1%), and *Bordetella parapertussis* (Table 1). 

The proportions of various respiratory tract pathogens that caused pneumonia in patients aged <18 years are shown in Figure 1A. Among patients aged <6 years, the most common respiratory tract pathogen was RSV, followed by rhinovirus, parainfluenza virus (1 + 2 + 3 + 4), adenovirus, metapneumovirus, *M. pneumoniae*, influenza A, and coronavirus. The dominant respiratory tract pathogens (according to percentage) were RSV (28%), rhinovirus (26%), adenovirus (14%), metapneumovirus (10%), parainfluenza virus type 3 (9%), parainfluenza virus type 4 (6%), parainfluenza virus type 1 (5%), *M. pneumoniae*, influenza A (2%), and coronavirus (1%). Among patients aged 6–17 years, the most common respiratory tract pathogens were rhinovirus, followed by *M. pneumoniae*, parainfluenza virus, adenovirus, and metapneumovirus (equal proportions), and then RSV and coronavirus (equal proportions). The major respiratory tract pathogens (according to percentage) were rhinovirus (17.9%), *M. pneumoniae* (14.3%), metapneumovirus (7.1%), adenovirus (7.1%), coronavirus (3.6%), RSV (3.6%), parainfluenza virus type 2 (3.6%), and parainfluenza virus type 4 (3.6%). 

The proportions of various respiratory tract pathogens that caused pneumonia in patients aged ≥18 years are shown in Figure 1B. Among patients aged 18–65 years, the most common respiratory tract pathogen was rhinovirus, followed by parainfluenza virus (1 + 2 + 3 + 4), coronavirus, RSV, and adenovirus. The major respiratory tract pathogens (according to percentage) were rhinovirus (6.6%), metapneumovirus (1.7%), coronavirus (1.4%), parainfluenza virus type 4 (1.4%), RSV (1.0%), adenovirus (0.7%), parainfluenza virus type 1 (0.3%), and parainfluenza virus type 3 (0.3%). Among patients aged >65 years, the most common pathogen was rhinovirus, followed by (in descending order) RSV, coronavirus, and metapneumovirus (equal proportions); adenovirus and parainfluenza virus (1 + 2 + 3 + 4) (equal proportions); and *B. parapertussis*. The major respiratory tract pathogens (according to percentage) were rhinovirus (4.4%), RSV (2.3%), coronavirus (2.2%), metapneumovirus (1.4%), adenovirus (0.7%), parainfluenza virus type 3 (0.4%), parainfluenza virus type 4 (0.3%), and *B. parapertussis* (0.1%).

The seasonal distribution of respiratory tract pathogens that caused pneumonia in patients aged <18 years is shown in Figure 2A. The common viral pathogen in spring was metapneumovirus followed by rhinovirus, in summer was adenovirus, in autumn was RSV followed by rhinovirus, and in winter was parainfluenza virus. The seasonal distribution of respiratory tract pathogen that caused pneumonia in patients aged ≥18 years is shown in Figure 2B. The common viral pathogen in spring and summer was rhinovirus, in autumn was coronavirus and RSV, and in winter was coronavirus followed by rhinovirus.

The proportions of various respiratory tract pathogens in the seasonal distribution in subgroup of patients aged <18 years were similar but a little different. Coronavirus among patients aged <6 years and among patients aged 6–17 years all occurred in autumn. Parainfluenza among patients aged <6 years and among patients aged 6–17 years did not occur in summer. Metapneumovirus among patients aged <6 years and among patients 6–17 years commonly occurred in spring. RSV among patients aged <6 years commonly occurred in autumn and could occur in all seasons, but among patients aged 6–17 years occurred in autumn. Adenovirus among patients aged <6 years occurred in summer and in all seasons, but among patients aged 6–17 years occurred in winter. The proportions of various respiratory tract pathogens in the seasonal distribution in subgroup of patients aged ≥18 years were a little similar. Coronavirus among patients aged 18–65 years and among patients aged >65 years did not occur in summer. Coronavirus in adults aged >65 years was most commonly detected in autumn. Metapneumovirus among patients aged 18–65 years and among patients aged >65 years commonly occurred in spring.

## 4. Discussion

We found that, from 2019 to 2021 in north Taiwan, rhinovirus, RSV, and parainfluenza were the three most common viral pathogens in patients with pneumonia. However, the most common pathogens slightly differed according to age. Among patients aged < 18 years, rhinovirus, RSV, and parainfluenza were the three most common viral pathogens; among patients aged ≥18 years, rhinovirus, RSV, and coronavirus were the three most common viral pathogens. However, the level of viral pathogen was high in children and low in adults. The third most common pathogen differed according to age. The distribution of viral pneumonia also differed between patients aged 0–6 years and patients aged 6–18 years; RSV and parainfluenza virus were dominant in patients aged 0–6 years. The distributions of viral pneumonia were generally similar between patients aged 18–65 years and patients aged >65 years, although RSV and coronavirus were more common in patients aged >65 years. Regarding seasonality, the most common viral pathogens that caused pneumonia were rhinovirus in spring, adenovirus and rhinovirus in summer, RSV and rhinovirus in autumn, and parainfluenza virus in winter. Among patients aged <18 years, the most common viral pathogens that caused pneumonia were metapneumovirus in spring, rhinovirus in summer, RSV in autumn, and parainfluenza virus in winter. Among patients aged ≥18 years, the most common viral pathogens that caused pneumonia were rhinovirus in spring and summer, RSV and coronavirus in autumn, and coronavirus in winter. Thus, there is a need to consider coronavirus as a cause of pneumonia (particularly during the COVID-19 pandemic). 

Coronavirus is an enveloped, single-stranded positive-sense RNA virus with a diameter of 60–140 nm and club-shaped spikes on its surface [8]. Viruses in the Coronaviridae family are grouped into four genera: alpha coronaviruses, beta coronaviruses, gamma coronaviruses, and delta coronaviruses [8]. Coronaviruses cause 5–10% of all upper respiratory tract infections in adults; they presumably cause severe respiratory infections in both children and adults [9]. Outbreaks of coronavirus respiratory infections usually occur in autumn or spring, but infections can occur in all seasons [9]. The severity and risk of infection are increased in adults with an underlying pulmonary disease, older patients, neonates, infants, young children, and immunocompromised hosts [10,11,12]. In our study, the percentages of pneumonia cases involving coronavirus were 1.9% overall, 1.6% in patients aged <18 years, and 2.0% in patients aged ≥18 years. The age group with the most cases of coronavirus-related pneumonia (3.6%) was patients aged 6–18 years (Figure 1A). Regarding seasonality, most cases of coronavirus-related pneumonia in children (aged <18 years) occurred in the autumn of 2019 (6.3%) and 2020 (3.4%). No cases were detected in the winter of 2019 and 2020, in the summer of 2020, or in the spring of 2020 and 2021. In contrast, most cases of coronavirus-related pneumonia in adults (aged ≥18 years) occurred in the winter of 2019 (11%), in the autumn of 2019 (3.7%), in the spring of 2021 (2.1%), and in the winter of 2020 (1.4%). No cases were detected in the spring or summer of 2020. Overall, most coronavirus-related cases occurred in the autumn of 2019 (6.3%). However, among patients aged >65 years, coronavirus-related pneumonia occurred most often in the autumn of 2020 (4.6%). For common coronavirus infections, supportive care with hand washing and careful treatment of body secretions is the main treatment/prevention strategy. However, for some coronaviruses that can cause acute respiratory distress syndrome (e.g., severe acute respiratory syndrome, Middle East respiratory syndrome, and COVID-19), hand washing and quarantine are recommended. Antiviral medications and vaccines have also been developed for COVID-19 [13].

RSV is an enveloped, non-segmented negative-strand RNA virus of the family *Pneumoviridae* family, Orthopneumovirus genus that can infect the respiratory tract [14]. It causes seasonal outbreaks of respiratory tract illness worldwide, usually in winter [15]. Although regional variation occurs, genotype A or B predominates in a single season, alternating annually [14]. Nearly all children aged >2 years have been infected with RSV, and reinfection is common. RSV can cause severe lower respiratory tract infections in high-risk groups, including infants (premature or not), patients with chronic lung or congenital heart disease, patients with Down syndrome, patients with persistent asthma, and immunocompromised patients [16]. RSV is a cause of death in older people, and is clinically clarified in adults admitted to hospitals with expanded use of molecular assays [14]. In our study, the overall percentage of RSV infection in patients with pneumonia was 4.3%, compared with 22.7% in patients with pneumonia who were aged <18 years and 2.0% in patients with pneumonia who were aged ≥18 years. The most common age of patients with RSV-related pneumonia was 0–6 years (28%; Figure 1A). Regarding seasonality, RSV-related pneumonia in children (aged <18 years) occurred most often in the autumn of 2019 (25%) and 2020 (44.8%), followed by the winter of 2019 (21.1%) and 2020 (18.5%), the spring of 2019 (12.5%) and 2020 (5.3%), and the summer of 2020 (10.0%). RSV-related pneumonia in adults (aged ≥18 years) occurred most often in the autumn of 2019 (3.7%), followed by the winter of 2020 (1.4%), the spring of 2021 (2.1%), and the winter of 2019 (11%). However, no cases were detected in the winter of 2019, spring of 2020, or summer of 2020. Among all patients, RSV infection occurred most often in the autumn of 2019 (25%). Among children aged 0–6 years, RSV infection occurred most often in the autumn of 2020 (50%). Preventive measures against RSV infection include hand washing; appropriate use of gloves, masks, gowns, and eye protection; and isolation. In high-risk groups, a humanised monoclonal antibody, palivizumab, can prevent hospitalisation [2,17]. Patients with a lower respiratory tract RSV infection primarily receive supportive treatment [17]. 

Adenovirus is a nonenveloped icosahedron with fibre-like projections and a double-stranded DNA genome [18]. It has a worldwide distribution, and infections can occur in all seasons. Adenovirus causes upper/lower respiratory infections, gastrointestinal tract infections, or conjunctivitis [18,19]. Most adenovirus infections are self-limiting, but severe pneumonia can occur [19]. Fatal infections can occur in immunocompromised patients; they occasionally occur in healthy children and adults [10,20]. In our study, the overall percentage of patients with adenovirus-related pneumonia was 2.0%; the percentage was 12.5% among patients aged <18 years and 0.7% among patients aged ≥18 years. The most common age of patients with adenovirus-related pneumonia was 0–6 years (14%; Figure 1A). Regarding seasonality, adenovirus-related pneumonia in children (aged <18 years) occurred most often in the winter of 2019 (31.6%) and in the summer of 2020 (20%), followed by the spring of 2019 (12.5%) and 2020 (15.8%), and in the autumn of 2019 (12.5%) and 2020 (3.4%). Most cases of adenovirus-related pneumonia in adults (aged ≥18 years) occurred in the spring of 2019 (2.5%), autumn of 2020 (1.6%), winter of 2020 (0.5%), and spring of 2021 (0.3%); no cases were detected in the winter of 2019 or summer of 2020. Overall, most cases of adenovirus-related pneumonia occurred in the winter of 2019 (21.4%). To prevent an adenovirus infection, individuals may use chlorine, formaldehyde, or heat to clean a contaminated environment or instruments. Some nucleotide analogues are used under temporary authorisation for severe adenovirus infections, including cidofovir and brincidofovir [21]. Live oral vaccines are highly efficacious in terms of reducing the risk of respiratory adenovirus infections, but they are not currently available in Taiwan [19].

Metapneumovirus is a member of the family Pneumoviridae, which includes large enveloped negative-sense RNA viruses [22]. Human metapneumovirus can cause upper and lower respiratory tract infections among patients in all age groups, but symptomatic disease most often occurs in young children or older adults [22]. Primary infection occurs before the age of 5 years, and humans are reinfected throughout life [23]. In our study, the overall percentage of metapneumovirus-related pneumonia cases was 2.4%; the percentage was 9.4% among patients aged <18 years and 1.5% among patients aged ≥18 years. The most common age of patients with metapneumovirus-related pneumonia was 0–6 years (10%; Figure 1A). Regarding seasonality, most cases of metapneumovirus-related pneumonia in children (aged <18 years) occurred in the spring of 2021 (47.4%) and 2020 (12.5%), followed by the winter of 2020 (7.4%). No cases were detected in the autumn of 2019 or 2020, the winter of 2019, or the summer of 2020. Most cases of metapneumovirus-related pneumonia in adults (aged ≥18 years) occurred in the spring of 2021 (3.4%) and the winter of 2020 (1.1%). No cases were detected in the winter of 2019, spring of 2020, summer of 2020, or autumn of 2019. Overall, most cases of metapneumovirus-related pneumonia occurred in the spring of 2021 (6.1%). Infection control measures include hand washing; appropriate use of gloves, masks, gowns, and eye protection; and isolation. Supportive care is the core component of treatment [24]. There are currently minimal clinical data available regarding antiviral therapies and vaccinations [24].

Rhinovirus is an RNA virus of the picornavirus family that causes >50% of upper respiratory tract infections in humans [25]. Rhinovirus is among the leading causes of viral bronchiolitis in infants, and it may play a role in asthma exacerbation [25]. In our study, the overall percentage of patients with rhinovirus-related pneumonia was 7.1%; the percentage was 24.2% among patients aged <18 years and 5.0% among patients aged ≥18 years. The most common age of patients with rhinovirus-related pneumonia was 0–6 years (26%; Figure 1A). Regarding seasonality, rhinovirus-related pneumonia in children (aged <18 years) occurred most often in the spring of 2021 (42.1%) and 2020 (12.5%), followed by the autumn of 2019 (31%) and 2020 (25%), the winter of 2020 (22.2%) and 2019 (10.5%), and the summer of 2020 (10%). Rhinovirus-related pneumonia in adults (aged ≥18 years) occurred most often in the winter of 2020 (5.9%), spring of 2021 (5.8%) and 2020 (2.5%), autumn of 2020 (3.1%), and summer of 2020 (2.1%). No cases were detected in the winter of 2019. Overall, rhinovirus-related pneumonia occurred most often in the autumn of 2019 (25%). Prevention measures include good hygiene. Pleconaril has been used for compassionate cases [2]. Supportive and symptomatic treatments are the core components of treatment; there is no effective vaccine available [26].

Influenza virus is a negative-sense, single-stranded RNA orthomyxovirus. There are three major types of influenza: A, B, and C [27]. Human influenza A and B viruses cause seasonal epidemics almost every winter, whereas influenza C virus less frequently causes respiratory illness. Classic influenza includes the sudden onset of fever, chills, and myalgias, followed by prominent upper respiratory tract symptoms [27]. However, younger children are less likely to exhibit a flu-like syndrome. Upper respiratory tract infection, laryngotracheitis (croup), bronchiolitis, and pneumonia are all possible presentations of influenza in younger children [2]. Gastrointestinal symptoms can be the primary symptoms in children [27]. Common complications of influenza include otitis media, pneumonia, and exacerbation of chronic pulmonary disease [28,29]. Severe influenza disease has greater risk for comorbidities including neurological, cardiac, metabolic, and haematological condition [29]. The risk of hospitalization or severe/complicated influenza is increased among younger children, as well as children with some underlying medical conditions [29]. Death most commonly occurs in high-risk groups, but it can also occur in healthy children [1,28,30]. In our study, the overall percentage of patients with influenza-related pneumonia was 0.2%; the percentage was 1.6% among patients aged <18 years and 0% among patients aged ≥18 years. The most common age of patients with influenza-related pneumonia was 0–6 years (2%; Figure 1A). Regarding seasonality, influenza-related pneumonia in children (aged <18 years) occurred only in the spring of 2019 (12.5%); no cases were detected in 2020 or 2021. Influenza-related pneumonia in adults (aged ≥18 years) was not detected. Influenza can be transmitted through respiratory droplets, aerosols, and contaminated objects or surfaces. To prevent transmission, frequent hand washing and covering of the mouth and nose when coughing or sneezing are recommended. Annual vaccination can help to provide protection against influenza. All persons aged ≥6 months should be vaccinated. Vaccination is effective in young children, including children aged <2 years. Influenza virus infections can be treated with supportive measures and (in severe cases) antiviral drugs, such as oseltamivir or zanamivir [31,32]. Treatment generally should be initiated within 48 h of symptom onset [30].

Parainfluenza viruses, members of the Paramyxoviridae family, are negative-sense, single-stranded, enveloped RNA viruses [33]. Parainfluenza viruses are a frequent cause of childhood disease, particularly among children aged <5 years, including pneumonia and laryngotracheobronchitis (croup) [30]. The clinical manifestation of parainfluenza viruses in hospitalized adults varies widely, immunocompromised and paediatric patients have an increased risk of severe infection and death [33]. In our study, the overall percentage of patients with parainfluenza-related pneumonia was 2.8%; the percentage was 17.2% among patients aged <18 years and 1.1% among patients aged ≥18 years. The most common age of patients with parainfluenza-related pneumonia was 0–6 years (20%; Figure 1A). Regarding seasonality, parainfluenza-related pneumonia in children (aged <18 years) occurred most often in the winter of 2019 (36.9%) and 2020 (30%), followed by the autumn of 2020 (10.3%) and 2019 (6.3%), and the spring of 2020 (5.3%); no cases were detected in the spring or summer of 2020. Most cases of parainfluenza-related pneumonia in adults (aged ≥18 years) occurred in the winter of 2020 (1.8%), spring of 2021 (0.6%), and autumn of 2020 (0.5%); no cases were detected in the winter of 2019, spring of 2020, or summer of 2020. Among all patients, parainfluenza virus was detected most commonly in the winter of 2019 (25%). Parainfluenza virus infections are usually self-limiting. Among patients with croup caused by parainfluenza, epinephrine and dexamethasone can reduce symptoms, resulting in shorter hospital stays and fewer return visits [30,34]. No licensed antiviral therapy is available for parainfluenza virus infections. However, aerosolised or systemic ribavirin with and without intravenous immunoglobulins has been used for the treatment of immunocompromised patients with severe disease [2,30]. There are no available vaccines, but several studies are ongoing. Parainfluenza viruses are easily transmitted through direct contact, as well as exposure to respiratory secretions and aerosols [30]. Thus, hand washing is an effective method to prevent transmission. 

During the COVID-19 pandemic, many vaccines were developed for SARS-CoV-2. COVID-19 vaccine efficacy data, including Pfizer–BioNTech (mRNA), Moderna (mRNA), Oxford-AstraZeneca (viral vector), Johnson & Johnson (viral vector), Gamaleya (Viral vector), Bharat Biotech (Viral vector), Sinovac Biotech (Inactivated virus), Sinopharm (Inactivated virus), and Novavax (Protein subunit) from phase III trials, were reported [35]. Now mRNA vaccine is most commonly applied to prevent COVID-19, and a third dose of a SARS-CoV-2 mRNA vaccine had high vaccine effectiveness against critical disease, hospitalization, and symptomatic infection in the pre-Omicron era [36]. Administered as 2 doses, mRNA vaccine is authorized and also expanded to children aged 5–11 years [37]. COVID-19 vaccine-related myocarditis was noted but with low overall incidence rates of 2–5 per million mRNA vaccines [38]. Influenza vaccines were already applied to people for many years. Quadrivalent influenza vaccine remains strongly recommended in high-risk populations [39]. However, active vaccines for RSV, rhinovirus, metapneuomoccus, parainfluenza, and adenovirus were not commonly applied. Nevertheless these pathogens were still common in people, especially children, during the COVID-19 pandemic. Developing vaccines to prevent these viral pneumonia may be considered for such high-risk groups.

## 5. Conclusions

During the COVID-19 pandemic (2019–2021), coronavirus in autumn in adults aged >65 years was most commonly detected, but was not detected in the summer in adults or children. RSV, rhinovirus, and adenovirus were detected in children aged 0–6 years in all seasons during this period. Among children aged 0–6 years, RSV was the most common viral pathogen, and RSV infection occurred most often in autumn. In both children and adults, metapneumovirus infection occurred most often in spring. However, no influenza virus-related pneumonia was detected in children or adults during 2020 or 2021. Additionally, the proportion of viral pneumonia cases was higher in children than in adults. Therefore, when children have viral pneumonia, pathogens other than SARS-CoV-2 should be considered. The COVID-19 pandemic period evoked a need for SARS-CoV-2 vaccination to prevent the severe complications of COVID. However, other viruses were also found. Vaccines for influenza were clinically applied. Active vaccines for other viral pathogens such as RSV, rhinovirus, metapneuomoccus, parainfluenza, and adenovirus may need to be developed for special groups in the future. 

## Figures and Tables

**Figure 1 vaccines-11-00905-f001:**
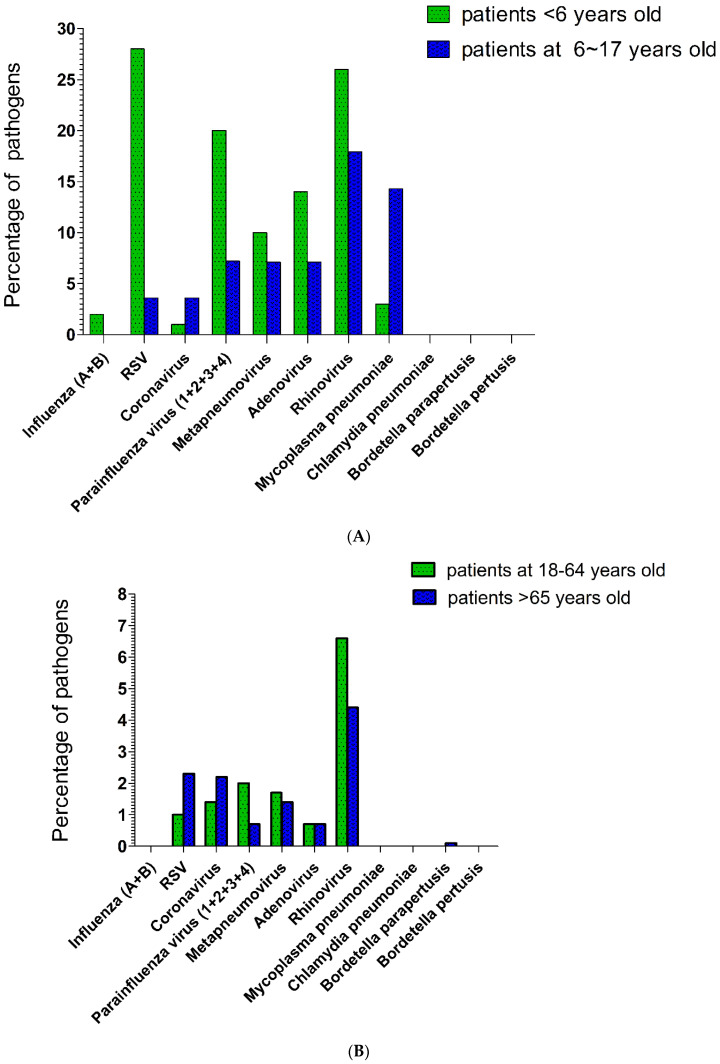
(**A**). The highest percentage of respiratory tract pathogens in pneumonia during the COVID-19 pandemic in the below-age-18 subgroup was RSV among patients <6 years old, and rhinovirus among patients 6–17 years old. (**B**). The highest percentage of respiratory tract pathogens in pneumonia during the COVID-19 pandemic in the above-age-18 subgroup was rhinovirus for both patients 18–64 years old and >65 years old.

**Figure 2 vaccines-11-00905-f002:**
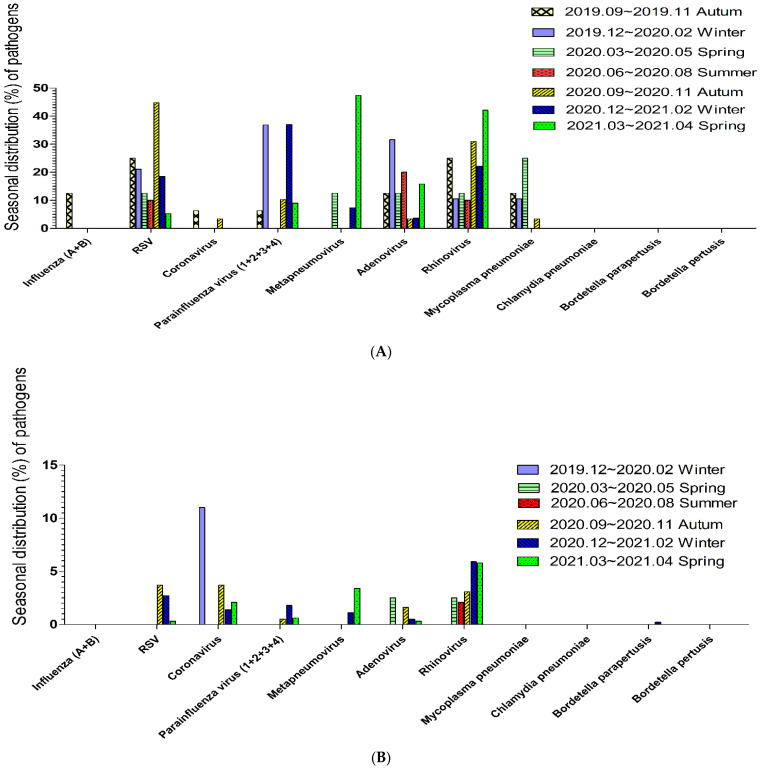
(**A**). The seasonal distribution of pathogens in pneumonia during the COVID-19 pandemic among patients below age 18; the common viral pathogen in spring was metapneumovirus, in summer was adenovirus, in autumn was RSV, and in winter was parainfluenza virus. (**B**). The seasonal distribution of pathogens in pneumonia during the COVID-19 pandemic among patients above age 18; the common viral pathogen in spring and summer was rhinovirus, in autumn was coronavirus and RSV, and in winter was coronavirus followed by rhinovirus.

**Table 1 vaccines-11-00905-t001:** Distribution of respiratory tract pathogens among patients.

Group	All	%	<18	%	>18	%
Number	1147	100.0	128	100.0	1019	100.0
Age (mean, sd)	63.40		3.65		70.90	
	(26.53)		(4.27)		(16.87)	
Male	761	66.3	77	60.2	684	67.1
Female	386	33.7	51	39.8	335	32.9
Influenza A	2	0.2	2	1.6	0	0.0
Influenza B	0	0.0	0	0.0	0	0.0
Influenza (total)	2	0.2	2	1.6	0	0
RSV	49	4.3	29	22.7	20	2.0
Coronavirus	22	1.9	2	1.6	20	2.0
Parainfluenza	6	0.5	5	3.9	1	0.1
virus 1						
Parainfluenza	1	0.1	1	0.8	0	0.0
virus 2						
Parainfluenza	13	1.1	9	7.0	4	0.4
virus 3						
Parainfluenza	13	1.1	7	5.5	6	0.6
virus 4						
Parainfluenza (total)	33	2.8	22	17.2	11	1.1
Metapneumovirus	27	2.4	12	9.4	15	1.5
Adenovirus	23	2.0	16	12.5	7	0.7
Rhinovirus	82	7.1	31	24.2	51	5.0
Mycoplasma pneumoniae	7	0.6	7	5.5	0	0.0
Chlamydia pneumoniae	0	0.0	0	0.0	0	0.0
Bordetella parapertusis	1	0.1	0	0.0	1	0.1
Bordetella pertusis	0	0.0	0	0.0	0	0.0

## Data Availability

Data is not available for sharing.

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
