# Peer review of "Viral Pneumonia during the COVID-19 Pandemic, 2019–2021 Evoking Needs for SARS-CoV-2 and Additional Vaccinations"

_vaccines, 2023, doi:10.3390/vaccines11050905_

Round 1

Reviewer 1 Report

The authors have presented an observational study data covering pneumonia cases from hospitalized patients during the COVID-19 pandemic from April 2019 to December 2021. Through this data the authors have highlighted the need of vaccination against RSV in younger (<6 years) and older age groups (and >65 years old patients) having weaker immune response. Further, they have also presented a distribution frequency of other respiratory pathogens among different age groups during different seasons in Taiwan. Most common pathogen was rhinovirus RSV, parainfluenza virus, adenovirus, metapneumovirus, Mycoplasma pneumonia, influenza virus and Coronavirus. The study provides an important data and would be useful to policy makers, clinicians and researchers involved in the development of vaccines and therapeutics to prioritize the areas of further development. Some of the specific comments are as follows: 

1.       A discussion in light of existing baseline data of disease burden by different pathogens, as available with relevant cross referencing, would be useful.   

2.       Line 107. The authors have mentioned that for both RNA and DNA extraction, Viral RNA minikit was used however, they should specify if the same kit can be use for both RNA and DNA extraction or they have used separate kit for DNA extraction.

3.       Line 98. Table 1. Title and data presentation can be improved. Correct title ‘The data of collected patient’ should be used. Suggestion – ‘Distribution of respiratory tract pathogens among patients’. It would be more presentable to use separate row instead of column to specify total proportions of specific groups e.g. Row 1. Influenza A, Row 2. Influenza B and Row 3. Total (influenza).

4.       Consider revising line 56.

5.       Data presentation in figures: Both Figure 1 and figure 2 – Proper title of Y axis should be used. Line graphs are typically used for presenting continuous time series data. The suggestion is to use histograms to present this data.  

6.      Grammatical and many typographical errors are present. Specifying units need to be fixed (e.g. Line Nos. 104, 109, 110, 112, 115). Similarly, presentation of temperature units needs to uniform (e.g. Line nos. 111, 124). Future tense is used in method section (2.2.2, line no 126 to 129). Extra space in line 54, 405, 406.

7. Provide details of FirmArray platform used for detection of viral pathogens. 

Author Response

Dear reviewers, Vaccines:

Thank you for your kind and professional review of our manuscript. The point by point reply is as following and red words in the manscript.

Reviewer 1

1.A discussion in light of existing baseline data of disease burden by different pathogens, as available with relevant cross referencing, would be useful.

Thanks reviewer’s comments.

We add some sentences

“RSV is a cause of death in older people, and its clinical clarify on adults admitted to hospital by expanded use of molecular assays [14].”

“Although regional variation occurs, genotypes A or B predominates in a single season, alternating annually [14].“

“RSV is a cause of death in older people, and its clinical clarify on adults admitted to hospital by expanded use of molecular assays [14].“

“The clinical manifestation of parainfluenza viruses in hospitalized adults varies widely, immunocompromised and the pediatric patients have an increased risk of severe infection and death [33].”

“Among patients with croup caused by parainfluenza, epinephrine and dexamethasone can reduce symptoms, resulting in shorter hospital stays and fewer return visits [30, 34].”

     “ COVID-19 vaccine-related myocarditis was noted but low overall incidence rates of 2-5 per million mRNA vaccines [38].”

  1. Line 107. The authors have mentioned that for both RNA and DNA extraction, Viral RNA minikit was used however, they should specify if the same kit can be use for both RNA and DNA extraction or they have used separate kit for DNA extraction.

Thanks reviewer’s comments.

Lyses the sample by agitation (bead beating) in addition to chemical lysis mediated by the Sample Buffer. Extracts and purifies all nucleic acids from the sample using magnetic bead technology.

We added sentence
“Operation process in the FilmArray system:
The push rod inside the instrument pushes the samples from the original position of the column into the detection bag. The samples were mechanically squeezed by the instrument and collided with the white ceramic beads that produced cell lysates and released nucleic acid substances. These nucleic acid substances were adsorbed by black magnetic beads that were located in the detection bag. And then the 1-3 slots in the bag are squeezed back and forth several times to purify the nucleic acid. The instrument injected buffer from the back-end column to extract the purified nucleic acid, and then entered the detection bag to do next-step PCR procedure.”

3.Line 98. Table 1. Title and data presentation can be improved. Correct title ‘The data of collected patient’ should be used. Suggestion – ‘Distribution of respiratory tract pathogens among patients’. It would be more presentable to use separate row instead of column to specify total proportions of specific groups e.g. Row 1. Influenza A, Row 2. Influenza B and Row 3. Total (influenza).

Thanks reviewer’s comments.

We changed table title “Table 1. Distribution of respiratory tract pathogens among patients.”

We separated row instead of column to specify total proportions of specific groups e.g. Row 1. Influenza A, Row 2. Influenza B and Row 3.  Influenza(total). 

Row 1. parainfluenza 1, Row 2. Parainfluenza 2, Row 3. Parainflenza 3, Row 4. Parainflenza 4 and  Row 5, Parainfluenza (total).

4.Consider revising line 56.

Thanks reviewer’s comments.

We revised line 56. Keywords: Viral pneumonia; COVID-19; SARS-CoV2; vaccination

  1. Data presentation in figures: Both Figure 1 and figure 2 – Proper title of Y axis should be used. Line graphs are typically used for presenting continuous time series data. The suggestion is to use histograms to present this data.Thanks reviewer’s comments.

We made changes and used histograms to present this data in both figure 1 and figure 2.

6 Grammatical and many typographical errors are present. Specifying units were be fixed (e.g. Line Nos. 104, 109, 110, 112, 115). Similarly, presentation of temperature units needs to uniform (e.g. Line nos. 111, 124). Future tense is used in method section (2.2.2, line no 126 to 129). Extra space in line 54, 405, 406.

We corrected units Line Nos. 104, 109, 110, 112, 115.

We uniform temperature units Line nos. 111, 124

We corrected with past tense in method section (2.2.2, line no 126 to 129).

We deleted extra space in line 54, 405, 406.

  1. Provide details of FirmArray platform used for detection of viral pathogens. 

Thanks reviewer’s comments.

Lyses the sample by agitation (bead beating) in addition to chemical lysis mediated by the Sample Buffer. Extracts and purifies all nucleic acids from the sample using magnetic bead technology.

We added sentence
“Operation process in the FilmArray system:
The push rod inside the instrument pushes the samples from the original position of the column into the detection bag. The samples were mechanically squeezed by the instrument and collided with the white ceramic beads that produced cell lysates and released nucleic acid substances. These nucleic acid substances were adsorbed by black magnetic beads that were located in the detection bag. And then the 1-3 slots in the bag are squeezed back and forth several times to purify the nucleic acid. The instrument injected buffer from the back-end column to extract the purified nucleic acid, and then entered the detection bag to do next-step PCR procedure.”

Reviewer 2 Report

In this manuscript “Coronavirus Infection as Important Cause for Viral Pneumonia 2 During the COVID-19 Pandemic, 2019–2021 Evoking Needs for 3 SARS-CoV2 and Additional Vaccinations in Taiwan”, Sheng-Chieh Lin et al. analysed the different cases of pneumonia in Taiwan in 2019-2021. The authors discussed then the need for a multivalent vaccine to prevent pneumonia caused by several respiratory viruses. This manuscript is a description of all these pneumonia cases in Taiwan.

Major comments:

1.     The manuscript contains a lot of repetitions. The results section is the description of the table 1 and the figures 1 and 2 present the same data already showed in the table 1.

2.     The table 1 is difficult to read and the figures 1 and 2 performed on excel are not good quality figures. As a suggestion for the authors, a representation of the data using single dots instead of averages would be more relevant for the figures 1 and 2.

3.     The authors should use more relevant references in the discussion section, especially regarding the brief description of each viruses.

Minor comments:

1.     Title: SARS-CoV-2 appears in the title but is not part of the data so the authors should reconsider the title to make it stick with the data.

2.     Line 94: How the authors excluded the patients with a previous SARS-CoV-2 infection?

3.     Table 1: The percentages are not always displayed and this is not that clear to understand for the Influenza and Parainfluenza lines. Moreover, the authors could discuss the high number of males compared to the females.

4.     Lines 104, 110 and many more: The symbol µ has been mistyped all long.

5.     Figure 1: The figure should have one title for part A and B. Moreover, the legend for the axe y is missing for the figure 1A.

6.     Line 241: RSV is not part of the Paramyxoviruses anymore but the Pneumoviridae.

7.     Lines 347 to 369: The authors should use proper references to describe hPIV and not only the number 28.

8.     Line 371 to 376: The authors should revise this part and use a more recent reference.

Author Response

Dear reviewers, Vaccines:

Thank you for your kind and professional review of our manuscript. The point by point reply is as following and red words in the manscript.

Reviewer 2

Major comments:

  1. The manuscript contains a lot of repetitions. The results section is the description of the table 1 and the figures 1 and 2 present the same data already showed in the table 1.

Thanks reviewer’s comments.

We made changes and used histograms to present this data in both figure 1 and figure 2 to made more clear understanding in the data of  subgroup.

  1. The table 1 is difficult to read and the figures 1 and 2 performed on excel are not good quality figures. As a suggestion for the authors, a representation of the data using single dots instead of averages would be more relevant for the figures 1 and 2.

Thanks reviewer’s comments.

We made changes and used histograms to present this data in both figure 1 and figure 2 to made more clear understanding in the data of subgroup.

  1. The authors should use more relevant references in the discussion section, especially regarding the brief description of each viruses.

     Thanks reviewer’s comments.

We added more relevant references in the discussion section   

  1. Nam HH. Ison MG. Respiratory syncytial virus infection in adults. BMJ, 2019. 366: p. L5021.
  2. Gill PJ., et al. Identification of children at risk of influenzarelated complications in primary and ambulatory care: a systematic review and metaanalysis. Lancet Respir Med, 2015.3(2):p. 139‐149.
  3. Russell E, Ison MG. Parainfluenza Virus in the Hospitalized Adult. Clin Infect Dis, 2017.65(9):p.1570-1576.
  4. Johnson D. Croup. BMJ Clin Evid, 2009.2009:p. 0321.
  5. Lee ASY., et al. Myocarditis Following COVID-19Vaccination: A Systematic Revie(Octo          ber2020-October 2021). Heart Lung Circ, 2022. 31(6):p. 757-765.

Minor comments:

  1. Title: SARS-CoV-2 appears in the title but is not part of the data so the authors should reconsider the title to make it stick with the data.

     Thanks reviewer’s comments.

     We changed our title “Viral Pneumonia During the COVID-19 Pandemic, 2019–2021 Evoking Needs for SARS-CoV2 and Additional Vaccinations. ”

  1. Line 94: How the authors excluded the patients with a previous SARS-CoV-2 infection?

     Thanks reviewer’s comments.

      In Taiwan, SARS-CoV-2 infection is a legal communicable disease. The suspicious people was detected by PCR. The confirmed people infected by SARS-CoV-2 was registered and had a record in NHI card. The medical staff could detect these patients’ data.

  1. Table 1: The percentages are not always displayed and this is not that clear to understand for the Influenza and Parainfluenza lines. Moreover, the authors could discuss the high number of males compared to the females.

     Thanks reviewer’s comments.

We made more clear and separated row instead of column to specify total proportions of specific groups e.g. Row 1. Influenza A, Row 2. Influenza B and Row 3.  Influenza(total). 

Row 1. parainfluenza 1, Row 2. Parainfluenza 2, Row 3. Parainflenza 3, Row 4. Parainflenza 4 and  Row 5, Parainfluenza (total).

  1. Lines 104, 110 and many more: The symbol µ has been mistyped all long.

     Thanks reviewer’s comments.

We corrected units Line Nos. 104, 109, 110, 112, 115.

  1. Figure 1: The figure should have one title for part A and B. Moreover, the legend for the axe y is missing for the figure 1A.

      Thanks reviewer’s comments.

We made changes and used histograms to present this data in both figure 1 and figure 2 to made more clear understanding in the data of  subgroup.

      We also added one title for part A and B.

  1. Line 241: RSV is not part of the Paramyxoviruses anymore but the Pneumoviridae.

      Thanks reviewer’s comments.

Respiratory syncytial virus belongs to the recently defined Pneumoviridae family, Orthopneumovirus genus.

We changed sentence “RSV is an enveloped, non-segmented negative-strand RNA virus of the family Pneumoviridae family, Orthopneumovirus genus that can infect the respiratory tract [14].”

  1. Lines 347 to 369: The authors should use proper references to describe hPIV and not only the number 28.

     Thanks reviewer’s comments.

We added more reference

33  Russell E, Ison MG. Parainfluenza Virus in the Hospitalized Adult. Clin Infect Dis, 2017.65(9):p.1570-1576.

  1. Johnson D. Croup. BMJ Clin Evid, 2009.2009:p. 0321.

  1. Line 371 to 376: The authors should revise this part and use a more recent reference.

       Thanks reviewer’s comments.

       We added sentence “COVID-19 vaccine-related myocarditis was noted but low overall incidence     rates of 2-5 per million mRNA vaccines [38].”

Round 2

Reviewer 2 Report

The authors have changed the references as needed and have modified the text to correct some mistakes. The new table has also been improved.

However, the figures still do not reach a good enough quality to be in a publication and the text does not add additional value to the figures. 

Author Response

Thanks reviewer’s comment 

We had made figure 2 clear.

Additionally, we have added contents of results for value.

Round 3

Reviewer 2 Report

The figures still do not reach a good enough quality to be in a publication. The authors should use a proper software.

Author Response

Thanks Academic Editor’s comments

1. We deleted the term % at the top of the y axis in figures 1A, 1B, 2A and 2B.

  1. We deleted the word The at the begining of the y axis title of figures 1A and 1B and the titles begin with the word Percentage: Percentage of pathogens

  1. We deleted the word The at the begining of the y axis title of figures 2A and 2B and include (%) in the title: Seasonal distribution (%) of pathogens
